# Global rewiring of cellular metabolism renders *Saccharomyces cerevisiae* Crabtree negative

Zongjie Dai[1,2], Mingtao Huang[1,2], Yun Chen[1,2], Verena Siewers[1,2] & Jens Nielsen [1,2,3,4]

*Saccharomyces cerevisiae* is a Crabtree-positive eukaryal model organism. It is believed that the Crabtree effect has evolved as a competition mechanism by allowing for rapid growth and production of ethanol at aerobic glucose excess conditions. This inherent property of yeast metabolism and the multiple mechanisms underlying it require a global rewiring of the entire metabolic network to abolish the Crabtree effect. Through rational engineering of pyruvate metabolism combined with adaptive laboratory evolution (ALE), we demonstrate that it is possible to obtain such a global rewiring and hereby turn *S. cerevisiae* into a Crabtree-negative yeast. Using integrated systems biology analysis, we identify that the global rewiring of cellular metabolism is accomplished through a mutation in the RNA polymerase II mediator complex, which is also observed in cancer cells expressing the Warburg effect.

[1] Department of Biology and Biological Engineering, Chalmers University of Technology, SE-41296 Gothenburg, Sweden. [2] Novo Nordisk Foundation Center for Biosustainability, Chalmers University of Technology, SE-41296 Gothenburg, Sweden. [3] Novo Nordisk Foundation Center for Biosustainability, Technical University of Denmark, DK-2800 Kgs. Lyngby, Denmark. [4] Beijing Advanced Innovation Center for Soft Matter Science and Engineering, Beijing University of Chemical Technology, Beijing 100029, China. Correspondence and requests for materials should be addressed to J.N. (email: nielsenj@chalmers.se)

The yeast *Saccharomyces cerevisiae* is a widely used model organism for studying the biology of eukaryal cells as well as it is extensively used as a cell factory for the production of pharmaceuticals, chemicals, and biofuels[1,2]. Its metabolism has evolved to have oxidative fermentation, meaning that even in the presence of oxygen, the yeast uses fermentative metabolism when glucose is in excess, a metabolic feature that is generally referred to as the Crabtree effect[3,4]. This million-year-old evolution feature ensures the advant.age in its ecological niche due to the ability to rapidly consume glucose and produce ethanol that has antiseptic properties. However, it generally results in reduced yields when this yeast is used as a cell factory. There is therefore much interest in rewiring the central carbon metabolism to abolish the Crabtree effect.

Eliminating pyruvate decarboxylase activity in yeast completely abolishes the Crabtree effect, but the growth deficiency of pyruvate decarboxylase minus (Pdc⁻) strains in excess glucose conditions[5] limits their application for biotechnology. Even though Pdc⁻ strains have been studied for last 25 years, only one strategy has so far enabled successful restoration of the growth of Pdc⁻ strains in a minimal medium with excess glucose. This strategy involves introducing *MTH1* mutations, which were originally identified from Pdc⁻ strains evolved to grow in excess glucose[6]. However, the specific growth rate of this strain was only at 0.1 h⁻¹, and acetyl-CoA generation in the cytosol relies on acetate supplementation and the native ATP-dependent acetyl-CoA synthetase[7].

To overcome this challenge, we create an alternative pyruvate dehydrogenase (PDH) bypass in *S. cerevisiae* with an ATP-independent acetyl-CoA synthesis pathway. With this, growth of a Pdc⁻ strain is successfully restored in minimal media with excess glucose. Combining rational design, adaptive laboratory evolution (ALE), and reverse engineering, the specific growth rate of the best strain reaches 0.218 h⁻¹, which is close to the maximum growth of *S. cerevisiae* with purely respiratory metabolism[8] and the maximum specific growth rate of most Crabtree-negative yeasts[9]. We find that, to unlock the millions of years of evolution that has determined metabolic features of *S. cerevisiae*, many different metabolic parts need to be engineered, and an important element is enabling global transcriptional alteration by having a mutation in the mediator complex that supports rewiring of cellular metabolism.

## Results

**Establishing a functional ethanol overflow negative yeast**. In *S. cerevisiae*, deletion of *PDC1, 5,* and *6* completely abolishes ethanol production. However, pyruvate decarboxylase is also an indispensable enzyme of the PDH bypass, which provides cytosolic acetyl-CoA required for lipid biosynthesis and hereby, for cell growth in a sugar-based media (Fig. 1a). This is the main reason for Pdc⁻ strains being growth deficient in the glucose media.

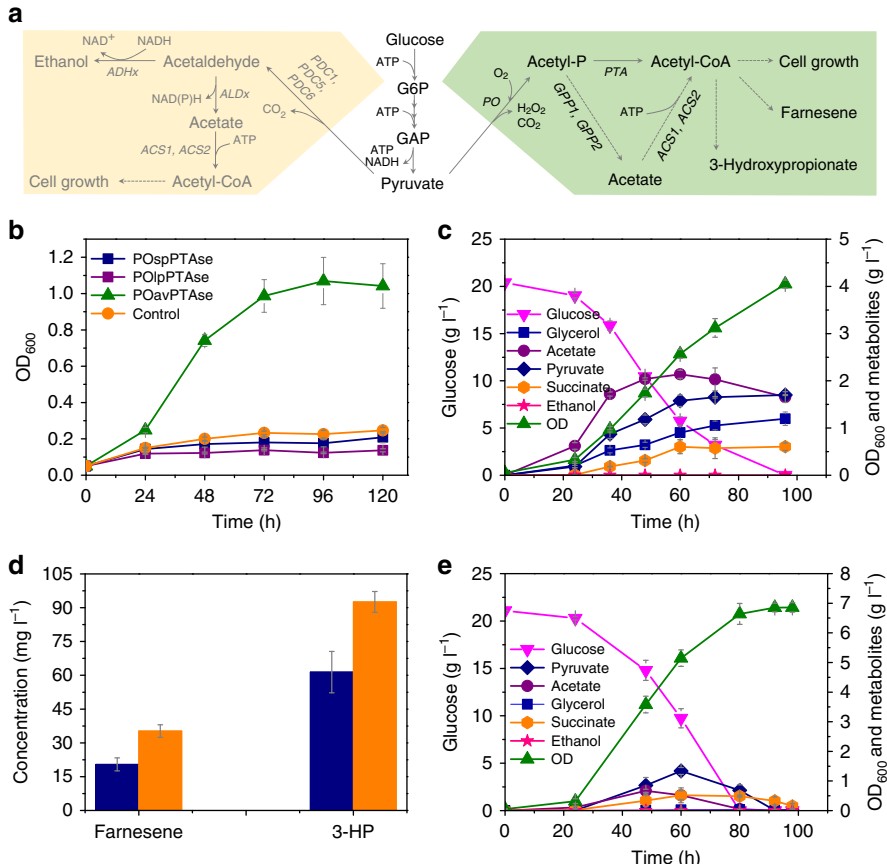

**Fig. 1** Establishment of a cytosolic acetyl-CoA synthetic pathway in Pdc⁻ *S. cerevisiae*. **a** The native and alternative cytosolic acetyl-CoA synthetic pathway in *S. cerevisiae*. The native metabolic network (yellow background) converts pyruvate to acetyl-CoA by pyruvate decarboxylase (Pdc1, 5, and 6), acetaldehyde dehydrogenase (Ald2, 3, 4, 5, and 6), and acetyl-CoA synthetase (Acs1 and 2). The metabolic network (green background) converts pyruvate to acetyl-CoA by pyruvate oxidase (PO) and phosphotransacetylase (PTA). **b** Growth curve of a Pdc⁻ yeast strain carrying different PO/PTA plasmids in a synthetic medium containing 20 g l⁻¹ glucose. sp: *S. pneumoniae*, lp: *L. plantarum*, av: *A. viridans*, se: *S. enterica*. **c** Growth and metabolite profiles of sZJD-11 (*acs2Δ::POav acs1Δ::PTAse*) in 20 g l⁻¹ glucose minimal medium. **d** Acetyl-CoA-derived farnesene and 3-hydroxypropanoate (3-HP) production in wild-type strain CEN.PK113-11C (orange bar) and sZJD-11 (blue bar) background strains. **e** Growth and metabolite profiles of sZJD-23 (*acs2Δ::POav acs1Δ::PTAse gpp1Δ gpp2Δ*) in 20 g l⁻¹ glucose minimal medium. All data represent the mean ± s.d. of biological triplicates

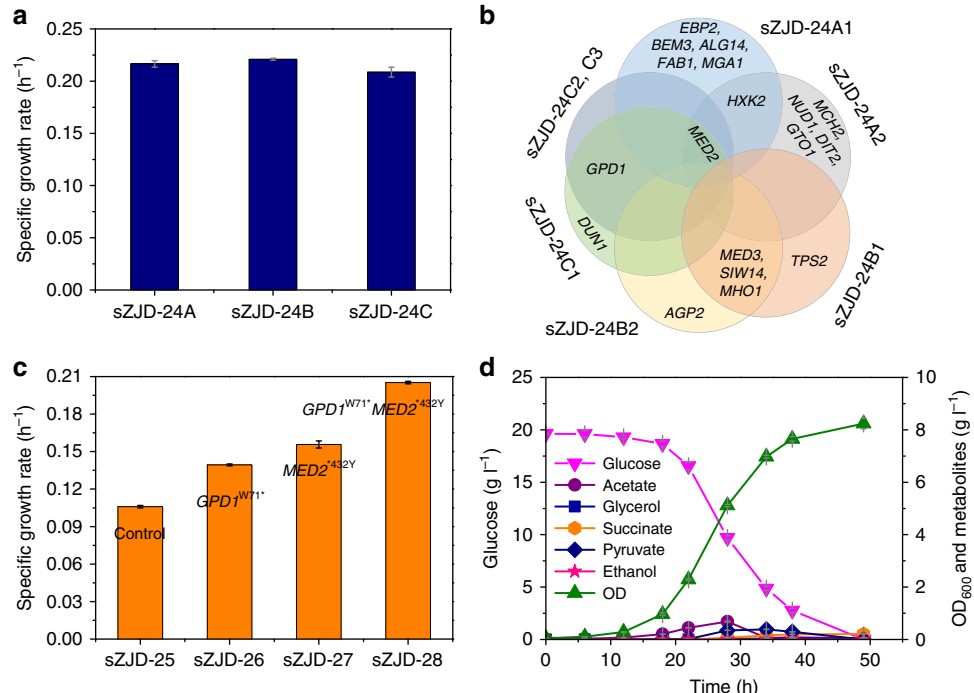

**Fig. 2** Characterization of adaptive laboratory evolution strains and reverse engineered strains. **a** Determination of the specific growth rate of evolved lines (sZJD-24A, sZJD-24B, and sZJD-24C) isolated at the end of the evolutionary experiment in 20 g l$^{-1}$ glucose minimal medium. **b** The genes with mutations identified in all of the isolated strains. sZJD-24A1 and A2 were from lines sZJD-24A, sZJD-24B1; B2 was from lines sZJD-24B, sZJD24C1; C2 and C3 were from line sZJD-24C. **c** Determination of the specific growth rate of reverse engineered strains based on sZJD-23 in shake flasks. **d** Growth and metabolite profiles of sZJD-28 in 20 g l$^{-1}$ glucose minimal medium. All data represent the mean ± s.d. of biological triplicates

To restore growth of a Pdc$^-$ strain in excess glucose media, we established an ATP-independent cytosolic acetyl-CoA producing pathway (Fig. 1a) in a Pdc$^-$ S. cerevisiae strain, which enabled growth in an excess glucose medium (Fig. 1b and Supplementary Figure 1). In this pathway, pyruvate oxidase (PO, EC 1.2.3.3) from *Aerococcus viridans* (av) catalyzes the decarboxylation of pyruvate to acetyl-phosphate, and subsequently, phosphotransacetylase (PTA, EC 2.3.1.8) from *Salmonella enterica* (se) converts acetyl-phosphate to acetyl-CoA (Fig. 1a), which was confirmed by the enzyme activity measurements (Supplementary Figure 2). To further verify the capacity of this pathway, the native acetyl-CoA synthetases, Acs1 and Acs2, were replaced by *PTAse* and *POav*, respectively, resulting in strain sZJD-11 (Supplementary Table 1), with a specific growth rate of 0.086 h$^{-1}$. The production of acetyl-CoA-derived chemicals farnesene (20.5 mg l$^{-1}$) and 3-hydroxypropionate (61.4 mg l$^{-1}$) in the sZJD-11 background strain (Fig. 1d) further demonstrated that this pathway is able to deliver cytosolic acetyl-CoA, even for heterologous pathways. To reduce acetate accumulation in sZJD-11 (Fig. 1c), glycerol 3-phosphate phosphatase (GPP) was identified as a promiscuous enzyme converting acetyl-phosphate into acetate (Supplementary Figure 3). The *GPP1* and *GPP2* double-deletion strain sZJD-23 accumulated less than 0.67 g l$^{-1}$ acetate in the culture medium, a decrease by 67.2% compared with sZJD-11. Other extracellular metabolites (pyruvate, succinate, and glycerol) were also produced at much lower levels than by sZJD-11. The specific growth rate of this strain reached 0.109 h$^{-1}$, which represents a 26.7% increase compared with sZJD-11 (Fig. 1e, c).

**Increasing growth of the Crabtree-negative S. cerevisiae.** Although the specific growth rate of this S. cerevisiae strain is similar to that of some natural Crabtree-negative yeasts, such as *Kluyveromyces nonfermentans* (0.101 h$^{-1}$) and *Eremothecium*

*sinecaudum* (0.117–0.122 h$^{-1}$), it is still much lower than for most natural Crabtree-negative yeasts (0.249–0.429 h$^{-1}$) (Supplementary Table 2).

We therefore established three independent yeast populations based on exposing sZJD-24 (prototrophic strain based on sZJD-23) (Supplementary Table 1) to ALE for 40 days, which is a duration compromising the selection of clones with improved fitness and not accumulating too many mutations. All of the three evolved populations have a higher specific growth rate compared with starting strain sZJD-24. (Supplementary Figure 4). The maximum specific growth rate of clones picked from each of these three populations reached 0.217 h$^{-1}$, 0.221 h$^{-1}$, and 0.209 h$^{-1}$, respectively (Fig. 2a). Through genome sequencing of seven clones (two each from sZJD-24A and sZJD-24B, three from sZJD-24C), we found a total of 19 single nucleotide variations (SNVs) in 18 genes (Supplementary Table 3). Although there were no shared mutations among all seven clones (Fig. 2b), a nonstop mutation in *MED2* was identified in five clones derived from lines sZJD-24A and sZJD-24C. The two clones of line sZJD-24B shared a mutation in *MED3*. Both Med2 and Med3 are components of the tail module of the RNA polymerase II mediator complex[10]. These results indicated that the mediator complex may play a key role in regulating cell growth. Only three SNVs were identified in the clones from line sZJD-24C, which shared a nonsense mutation in *GPD1* encoding a NADH-dependent glycerol-3-phosphate dehydrogenase[11]. For all clones in line sZJD-24B, besides *MED3*, we found shared mutations in *SIW14* and *MHO1*. Siw14 is tyrosine phosphatase involved in actin organization and endocytosis[12], Mho1 is a protein of unknown function.

Therefore, *MED2*, *MED3*, *GPD1*, *HXK2*, and *SIW14* were chosen as reverse engineering targets to evaluate if mutations in these gene were causal. We successfully obtained the *GPD1*$^{W71*}$ and *MED2*$^{*432Y}$ single mutant strains sZJD-26 and sZJD-27, respectively, using the Cas9-expressing strain sZJD-25. The

**Table 1 Physiological parameters of the wild-type and engineered strains[a]**

| Parameter | CEN.PK113-11c | sZJD-25 | sZJD-28 |
|---|---|---|---|
| $\mu$ ($h^{-1}$) | 0.374 ± 0.013 | 0.140 ± 0.003 | 0.218 ± 0.006 |
| $Y_{biomass}$ (g g$^{-1}$) | 0.126 ± 0.000 | 0.306 ± 0.003 | 0.368 ± 0.026 |
| $q_{glu}$ (mmol g$_{DW}$$^{-1}$ h$^{-1}$) | −16.405 ± 0.587 | −2.284 ± 0.026 | −3.255 ± 0.264 |
| $q_{eth}$ (mmol g$_{DW}$$^{-1}$ h$^{-1}$) | 23.694 ± 0.485 | ND | ND |
| $q_{gly}$ (mmol g$_{DW}$$^{-1}$ h$^{-1}$) | 1.964 ± 0.061 | ND | ND |
| $q_{ace}$ (mmol g$_{DW}$$^{-1}$ h$^{-1}$) | 0.361 ± 0.015 | 0.810 ± 0.017 | 1.397 ± 0.265 |
| $q_{pyr}$ (mmol g$_{DW}$$^{-1}$ h$^{-1}$) | 0.152 ± 0.003 | 0.026 ± 0.001 | 0.047 ± 0.014 |
| $q_{CO2}$ (mmol g$_{DW}$$^{-1}$ h$^{-1}$) | 27.461 ± 0.877 | 5.461 ± 0.043 | 7.633 ± 0.869 |
| $q_{O2}$ (mmol g$_{DW}$$^{-1}$ h$^{-1}$) | 3.575 ± 0.032 | 4.319 ± 0.021 | 6.400 ± 0.704 |
| RQ | 7.683 ± 0.314 | 1.264 ± 0.016 | 1.192 ± 0.010 |

[a]Data are shown mean values ± standard deviations of triplicates
ND not detected

specific growth rate of these two strains increased by 31.5 and 47.0% compared with the starting strain sZJD-25, reaching 0.139 h$^{-1}$, and 0.156 h$^{-1}$ respectively. A MED2$^{*432Y}$ and GPD1$^{W71*}$ double-mutant strain sZJD-28 reached an even higher specific growth rate of 0.205 h$^{-1}$, which is 98% of the specific growth rate of the evolved line sZJD-24C (Fig. 2c), showing a clear causal effect of these two mutations. This is consistent with the finding that these two mutations were the only two SNVs found in the evolved strains sZJD-24C2 and sZJD-24C3 (Fig. 2b). sZJD-28 consumed glucose with faster rate and reached higher OD$_{600}$ value compared with starting strain sZJD-23. The extracellular metabolites were also lower than those of sZJD-23 (Figs. 2d, 1e). Compared with wild-type strain CEN.PK113-11c, sZJD-28 had much higher biomass yield and lower RQ value, which is close to 1. The maximal specific growth rate of sZJD-28 reached 0.218 h$^{-1}$ (Table 1). Thus, through identification of targets using ALE, we managed to engineer a better and faster growing Crabtree-negative S. cerevisiae with reduced carbon loss.

**Transcriptional profiles of the Crabtree-negative strain**. To understand the underlying mechanism of how the mutations results in faster growth rate of the reverse engineered strains, we used RNA-Seq to perform transcriptome analysis of the parental strain sZJD-25 and the three strains sZJD-26, sZJD-27, and sZJD-28.

Transcriptome analysis showed that in sZJD-28, totally, 2096 genes, about 33% of all genes of S. cerevisiae, were significantly (padj < 0.01) differentially expressed and 1562 genes in sZJD-27 compared with sZJD-25 (Fig. 3a). This indicated that the nonstop mutation of MED2 resulted in a global impact on the metabolic network. Med2 is one of the subunits of the tail module[10], which is one of the four parts of the mediator complex, and it is required for the regulated transcription of nearly all RNA polymerase II-dependent genes in S. cerevisiae[13–15]. The tail module mediates mediator complex-associated transcriptional regulation on SAGA-regulated, TATA-containing genes which account for about 15% of all the genes in yeast[16,17] (Supplementary Figure 5). Indeed, almost half of all TATA-containing genes had significantly altered expression in the MED2 mutant strains sZJD-28 (7.5%) and sZJD-27 (6.8%) (Fig. 3b). It indicated a clear causal effect of this mutation. Additionally, the expession level of TATA-containing genes in the MED2 mutant strain were predominantly downregulated, especially in the double-mutant strain sZJD-28 (Fig. 3c), which is clearly seen in the Volcano plot of sZJD-28 (Supplementary Figure 6). GO Slim Mapper analysis on the 114 shared TATA-containing genes showed that GO terms related to response to cellular amino acid metabolic processes, carbon metabolism, and chemical and oxidative stress were enriched (Supplementary Data 1). It indicated that altering the

expression of genes with these GO terms may play an indispensable role in improving the cell growth rate.

The reporter GO term analysis showed that genes associated with GO terms related to translation are upregulated (upper part of heat map in Fig. 3d), whereas genes associated with GO terms related to carbon metabolism are downregulated (lower part of heat map in Fig. 3d). Protein synthesis is required for cell growth and needs ribosomes to polymerize amino acids into polypeptide chains. The cellular growth rate is linearly correlated with ribosome abundance, and the expression of ribosome-associated genes, including genes coding for ribosomal proteins and rRNA biogenesis, therefore affects the growth rate[18,19]. Indeed, in sZJD-28 and sZJD-27, protein synthesis-associated genes were significantly upregulated (Fig. 3d and Supplementary Figure 7). In addition, reporter TFs analysis showed that genes controlled by chromatin remodeling-related TFs such as Snf2, Snf6, Sin3, Sas3, and Rsc1 were significantly changed in sZJD-27 and sZJD-28, in contrast to sZJD-25 (Supplementary Figure 8-9). It suggested that gene transcription may have become more active, supporting increased protein synthesis in these two strains. However, increasing the ribosomal protein fraction would reduce the fraction of metabolic proteins[20]. Indeed, GO terms related to carbon metabolism-contained genes were significantly downregulated in sZJD-27 and sZJD-28 (Fig. 3d and Supplementary Figure 7). Downregulating of these metabolic genes would save the resource for ribosomal proteins synthesis, as glycolytic enzymes account for a major fraction of the cellular proteome. These results suggested that the introduced mutations led to redistributed and active protein synthesis, which may support a faster cell growth rate.

## Disscussion

Pdc$^-$ S. cerevisiae strains cannot grow in batch cultures on synthetic glucose medium. Two reasons are lacking of cytosolic acetyl-CoA supply and limited capacity of reoxidation of cytosolic NADH[21]. In sZJD-25, the PO/PTA pathway can produce acetyl-CoA in the cytosol, which supported the growth of Pdc$^-$ S. cerevisiae strains in an excess glucose medium. However, the reoxidation of cytosolic NADH still mainly relies on the mitochondrial respiratory chain due to the absence of alcoholic fermentation. In sZJD-25, the unrestricted glucose uptake and high glycolytic activity would particularly cause problems with recycling of cytosolic NADH to NAD$^+$. Compared with sZJD-25, the high-affinity glucose transporter genes HXT2, HXT4, HXT6, HXT7, HXT10, and HXT14 were upregulated and the low/medium-affinity glucose transporter genes HXT1, HXT3, HXT5, HXT9, and HXT11 were downregulated in sZJD-28. (Fig. 3e). The fold changes of high-affinity glucose transporter genes were higher than that of the low-affinity ones (Supplementary

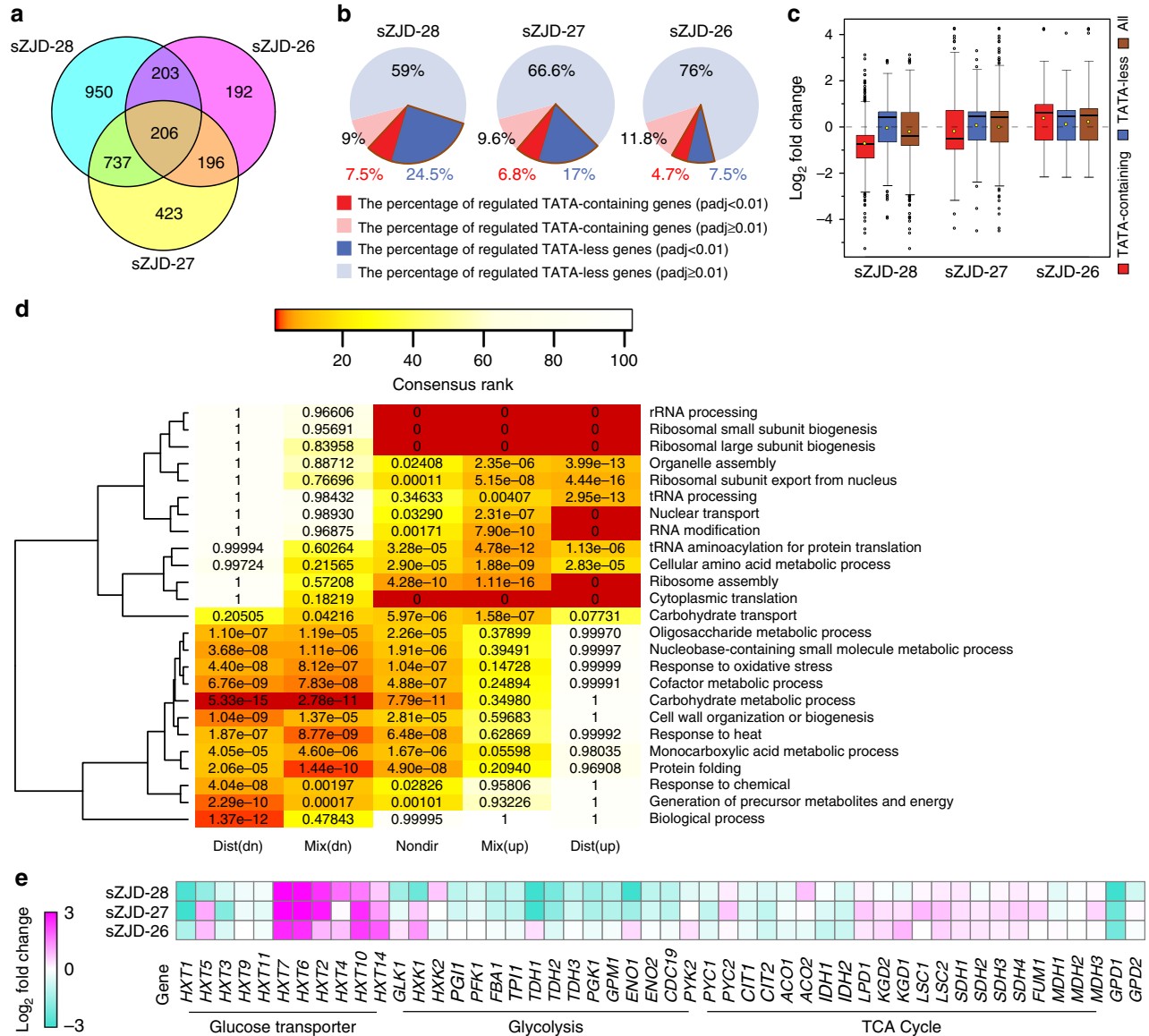

**Fig. 3** Transcriptional analysis of reverse engineered strains and parental strain. **a** Venn diagram showing the numbers of significantly (padj < 0.01) regulated genes of reverse engineered strains compared with the parental strain. **b** Fraction of TATA-containing and TATA-less genes among differentially regulated genes in sZJD-28, sZJD-27, and sZJD-26. **c** Box plot of the log$_2$-fold change of the significantly (padj < 0.01) regulated genes in sZJD-28, sZJD-27, and sZJD-26. The bold line in the box represents the median and the yellow dot represents the mean. The lower and upper bounds of the box indicate the first and third quartiles, respectively, and whishers represent ±1.5× the interquartile range (IQR). **d** Reporter GO term analysis of the transcription profiles of sZJD-28. The color key shows the rank of the GO terms, and the significance (*p* value) of the GO terms is included in each cell of the heatmap. GO terms that have a consensus rank ≤10 in any of the groups are shown in the heatmap. Dist (dn) is distinct-directional down, Mix (dn) is mixed-directional down, Nondir is non-directional change, Mix (up) is mix-directional up, and Dist (up) is distinct-directional up. **e** Transcriptional levels of genes related to glucose transportation, glycolysis pathway, and TCA cycle in sZJD-28, sZJD-27 and sZJD-26 compared with sZJD-25

Figure 10). Expression of high-affinity glucose transporters were repressed by high levels of glucose and induced by low levels of glucose[22]. Transcriptional changes in these glucose transporters indicated that glucose uptake was restricted in sZJD-28 compared with sZJD-25. Additonally, almost all of the genes in glycolysis were downregulated in sZJD-28 (Fig. 3e). Restricted glucose uptake and activity of glycolysis would release the burden on the respiration chain, which is consistent with the the expression profiles of oxidative phosphorylation (OXPHOS) genes, especially in sZJD-28 NADH dehydrogenase genes (*NDE1* and *NDE2*) and ubiquinol-cytochrome c oxidoreductase genes (complex III), which were significantly downregulated compared with sZJD-25 (Supplementary Figure 11). These results indicate that the limited

glucose consumption capacity of sZJD-25, which was enhanced by the introduced mutations led to higher glycolytic flux in sZJD-28, but now with balancing of NADH formation and oxidation back to NAD+. The faster glucose consumption rate, higher biomass yield and lower RQ of sZJD-28 compared with that of sZJD-25 (Table 1) further confirmed the higher glycolytic flux and respiration rate in the reverse engineered sZJD-28. The balance between fermentation and respiration may lead to efficient carbon and electron flux in the cell, which can support the faster growth rate.

Taken together, we demonstrated that the Crabtree-positive *S. cerevisiae* can be turned into a Crabtree-negative yeast by systematic engineering, which included rational pathway design and

system biology analysis. The growth rate of this engineered *S. cerevisiae* reached 0.218 h$^{-1}$, which was two folds of previous *MTH1* reverse engineered strains and almost reached the level of many natural Crabtree-negative yeasts. By systems biology analysis, the mediator complex was identified as a global regulator involved in rewiring the central carbon metabolism and allocating protein synthesis in a way that favors faster growth of this Crabtree-negative *S. cerevisiae*. We believe that the derived yeast strain represents a possible platform strain for use in biotechnology as well as global rewiring of yeast metabolism through engineering; the mediator complex may be used as a strategy in metabolic engineering of yeast. Additionally, our finding on restricting glucose flux by modulation of the conservative mediator complex may give an insight into cancer metabolism due to the similarity between Crabtree effect and Warburg effect in cancer cells[23]. Thus, many cancer cells have altered pyruvate metabolism[24] and mutations in the mediator complex[25,26].

## Methods

**Strains and plasmids.** The yeast *S. cerevisiae* CEN.PK YMZ-E1, IMI076, and CEN.PK 113-11C were used as host strain for strain engineering. The strains, plasmids, and primers used in this study are listed in the Supplementary Tables 1, 4 and Supplementary Data 2, respectively. POav, POsp, POlp, and PTAse (Supplementary Data 2) were codon-optimized for yeast expression and synthesized by Genscript.

**Reagents.** Primers were synthesized by Sigma-Aldrich. DNA purification and plasmid extraction kits, Taq DNA polymerase, and restriction enzymes were the products of ThemoFisher Scientific. PrimeStar DNA polymerase was purchased from TaKaRa Bio. For genomic DNA extraction, the Blood & Cell culture DNA Kit (Qiagen) was used. For RNA extraction, the RNeasy Mini Kit (Qiagen) was used. All chemicals used were purchased from Sigma-Aldrich, if not otherwise stated.

**Culture conditions and media.** All *S. cerevisiae* strains were cultivated at 200 r.p. m., 30°C. To determine the growth rate, strains were inoculated into the medium with an initial OD$_{600}$ of 0.05. YPD or YPE media (10 g l$^{-1}$ yeast extract, 20 g l$^{-1}$ peptone, 20 g l$^{-1}$ glucose or ethanol) were used for preparing competent cells. Synthetic complete media without uracil (SCD-URA or SCE-URA) were used to grow strains containing *URA3*-based plasmids. The media consisted of 6.9 g l$^{-1}$ yeast nitrogen base (YNB) without amino acids (Formedium), 770 mg l$^{-1}$ complete supplement mixture (CSM, w/o uracil) (Formedium), and 20 g l$^{-1}$ glucose or ethanol. SCD+ 5'-FOA or SCE+ 5'-FOA plates were used to select against the *URA3* marker and which contained 6.9 g l$^{-1}$ YNB, 770 mg l$^{-1}$ CSM, 0.8 g l$^{-1}$ 5-fluoroorotic acid, and 20 g l$^{-1}$ glucose or ethanol. Stains containing a *KanMX* cassette were selected on YPD plates containing 200 mg l$^{-1}$ G418. Shake flask batch fermentation was carried out in minimal medium, pH 6.3, consisting of 7.5 g l$^{-1}$ (NH$_4$)$_2$SO$_4$, 14.4 g l$^{-1}$ KH$_2$PO$_4$, 0.5 g l$^{-1}$ MgSO$_4$·7H$_2$O, 20 g l$^{-1}$ glucose, trace metal, and vitamin solution[27] supplemented with 40 mg l$^{-1}$ uracil and/or 40 mg l$^{-1}$ histidine, if needed. For batch fermentation in bioreactors, the methods in previous work[28] were followed. Specifically, strains were grown in defined minimal medium containing: 20 g l$^{-1}$ glucose, 5 g l$^{-1}$ (NH$_4$)$_2$SO$_4$, 3 g l$^{-1}$ KH$_2$PO$_4$, 0.5 g l$^{-1}$ MgSO$_4$·7H$_2$O, 125 μl antifoam 204 (Sigma-Aldrich, USA), 40 mg l$^{-1}$ uracil, 40 mg l$^{-1}$ histidine, trace metals solution, and vitamins. Batch fermentations were performed at 30°C in 1.2 l bioreactors (DasGip, Germany) with a working volume of 500 ml. Cultures were operated with 800 r.p.m. agitation and 1vvm gas flow of pure dried air. Culture pH was maintained at 5.0 by automated addition of 2 M KOH or 2 M H$_2$SO$_4$. The CO$_2$ and O$_2$ concentrations in the exhaust gas were analyzed real time with a GA4 gas analyzer (DasGip, Germany). The oxygen uptake rate at each time point was calculated by correcting for the dilution caused by CO$_2$ production according to the Eq. 1:

$$OUR = \frac{F \times O_2 in}{22.4} - \frac{N_2 in}{N_2 out} \times \frac{F \times O_2 out}{22.4} \qquad (1)$$

where, *F* is the gas flow existing the bioreactor, O$_2$ in is the fractional concentration of O$_2$ in the air entering the bioreactor, O$_2$ out is the fractional concentration of O$_2$ in the gas existing the bioreactor, N$_2$in is the combined fractional concentration of nitrogen and argon in the air entering the bioreactor (0.791) and N$_2$ out is the combined fractional concentration of nitrogen and argon existing the bioreactor (1-O$_2$out-CO$_2$out). The total mols of oxygen consumed until each time point were calculated by integrating the OUR curve against time, and the resulting values were against the biomass concentration in grams of cell dry weight. The specific oxygen consumption rate was calculated by taking the slope of the linear trendline and multiplying by the growth rate. For CO$_2$ calculation, first the release rate was

calculated by Eq. 2:

$$CER = \frac{F \times CO_2 out}{22.4} - \frac{F \times CO_2 in}{22.4} \qquad (2)$$

and then, it is the same procedure as for O$_2$.

**Genetic manipulation.** For plasmid construction, standard procedures of restriction enzyme digestion and ligation were used[29]. For gene deletion, integration, and point mutation, two strategies were used in this study. One way is seamless gene deletion or integration by using *Kluyveromyces lactis URA3* (*KlURA3*) as a selection marker[30]. The *KlURA3* marker was flanked by two homologous sequences, which were used for looping out the *KlURA3* marker after gene deletion and integrating via yeast homologous recombination system. Removing the *KlURA3* marker from the strain can be performed by using selective SCD+ 5'-FOA or SCE+ 5'-FOA plates. The other way for gene deletion, integration, and point mutation is CRISPR–Cas9-based strategy[31]. The Cas9 expression cassette was first fused with a *KanMX* cassette and integrated into *X-2* site of the yeast genome between cycles of CRISPR–Cas9 system. For ALE, *URA3* and *HIS3* were integrated into *XI-5* and *X-2* sites of the starting strain sZJD-23, respectively, resulting in strain sZJD-24. All of the plasmids and gene deletion or integration cassettes were transformed into yeast strains by using the LiAc/SS DNA/PEG method[32].

**Metabolite extract and analysis.** For HPLC analysis, 1 ml of the fermentation broth was centrifuged at 18,000 g for 5 min at 4°C. The supernatant was filtered through a 0.2 μm filter (VWD) then analyzed on an Aminex HPX-87H column (Bio-rad) in an Ultimate 3000 HPLC system (ThermoFisher Scientific). To detect extracellular glucose, ethanol, glycerol, acetate, pyruvate, and succinate, the column was set at 45°C and eluted with 5 mM H$_2$SO$_4$ at a flow rate of 0.6 ml min$^{-1}$. For 3-HP analysis, the column was set at 60°C and eluted with 1 mM H$_2$SO$_4$ at a flow rate of 0.6 ml min$^{-1}$ [33]. For farnesene analysis, procedures and conditions in the previous work by Tippmann et al[34]. were followed. Specifically, a sample was taken from the cultivation broth after 72 hours. An aliquot was taken from the dodecane phase and diluted five times in hexane for injection in GC-FID. The 2-μl samples were injected in splitless mode, with injection temperature at 200°C. ZB-50 column (30 m × 0.25 mm I.D., 0.25 μm film thickness; Phenomenex, Torrance, CA, USA), were used with helium as carrier gas at a flow rate of 1 ml min$^{-1}$. Initial oven temperature was set at 50°C for 1.5 min, increased up to 170°C (30°C min$^{-1}$) and held for 1.5 min. The temperature was then increased to 300°C (15°C min$^{-1}$) and maintained for 3 min.

**Enzyme activity assay.** For cell extract preparation, yeast cells were harvested during the exponential phase and washed with cold enzyme activity detection buffer (1 M potassium phosphate, pH 6.7 and 100 mM Tris HCl, pH 7.4 for pyruvate oxidase and phosphotransacetylase, respectively). Then, the cell suspension was transferred into 2 ml lysing matrix tubes containing beads (MP Biomedicals). Cells were lysed by the MP FastPrep-24 instrument with four 20 sec cycles at the speed of 4.0 m s$^{-1}$; the cell lysate was cooled down in ice for 5 min between cycles. Cell debris was removed from the cell lysate by centrifugation at 18,000 g for 5 min at 4°C, and the supernatant was used for enzyme activity assays. Protein concentration was determined using the Quick Start Bradford Protein Assay (Bio-Rad). For pyruvate oxidase and phosphotransacetylase activity analysis, previously described methods were used[35,36]. Specifically, for pyruvate oxidase, spectrophotometric stop rate determination method was used. The assay condition was: 37 °C, pH = 6.7, 565 nm, and light path = 1 cm. Reagent A: 1 M potassium phosphate buffer, pH 6.7 at 37°C, adjust to pH 6.7 with 1 M NaOH; Reagent B: 1 mM flavin adenine dinucleotide solution, prepare fresh; Reagent C: 10 mM thiamine pyrophosphate solution, prepare fresh; Reagent D: 15 mM 4-aminoantipyrine solution; Reagent E: peroxidase enzyme solution containing 50 purpurogallin units ml$^{-1}$ in deionized water using peroxidase, immediately before use; Reagent F: 100 mM MgCl$_2$; Reagent G: 0.2% (v/v) N,N-dimethylaniline suspension; Reagent H: prepare by combining 1 ml of Reagent F and 2 ml of Reagent G, prepare fresh; Reagent I: 1 M sodium pyruvate solution; Reagent J: 200 mM sodium phosphate solution; Reagent K: 100 mM citric acid solution; Reagent L: McIlvain Buffer, prepare 100 ml by adding 63.15 ml of Reagent J and 36.85 ml of Reagent K; Reagent M: 100 mM ethylenediaminetetraacetic acid solution, adjust to pH 5.5 with either 1 M NaOH or 1 M HCl; Reagent N: 10 mM potassium phosphate with 10 μM flavin adenine dinucleotide solution, pH 7.0. Procedure: prepare a reaction cocktail by pipetting (in ml) the following reagents into a suitable container: deionized water 1.70, Reagent A 2.00, Reagent B 0.10, Reagent C 0.20, Reagent D 1.00, and Reagent E 1.00. Mix by swirling and equilibrate to 37°C. Pipette (in ml) the following reagents into suitable containers (test and blank): reaction cocktail 0.60, Reagent H 0.30, and Reagent I 0.10. Mix by inversion and equilibrate to 37°C. Monitor the A565 nm until constant, then add: Reagent N 0.02 (blank) and Reagent O (test), immediately mix by inversion and incubate at 37°C for exactly 10 min. Then add Reagent M 2.00 in the both test and blank. Mix by inversion and incubate at 25°C for exactly 5 min. Transfer to suitable cuvettes and record at 565 nm. For phosphotransacetylase, continuous spectrophotometric rate determination method was used. The assay condition was: 25°C, pH = 7.4, 233 nm, and light path = 1 cm. Reagent A: 100 mM Tris HCl Buffer, pH 7.4 at 25°C, adjust to pH 7.4 at 25°C with

5 M HCl; Reagent B: 100 mM glutathione solution prepare fresh; Reagent C: 6.5 mM coenzyme A solution, prepare fresh; Reagent D: 220 mM acetyl phosphate solution, prepare fresh; Reagent E: 1 M ammonium sulfate solution; Reagent F: 25 mM Tris HCl Buffer with 500 mM ammonium sulfate, pH 8.0 at 25°C, adjust to pH 8.0 at 25°C with 1 M HCl. Procedure: pipette (in ml) the following reagents into UV cuvettes (test and blank): Reagent A 2.60, Reagent B 0.05, Reagent C 0.20, Reagent D 0.10, and Reagent E 0.03. Mix by inversion and equilibrate to 25°C. Moniter A233 nm until constant, then add: Reagent G 0.02 (test) and enzyme 0.02 (blank), immediately mix by inversion and record the increase in A233 nm for approximately 5 min. Obtain the $\Delta A233$ nm min$^{-1}$ using the maximum linear rate for both the test and blank.

**Adaptive laboratory evolution**. Adaptive laboratory evolution experiments with *S. cerevisiae* sZJD-24 were carried out by serial dilutions in shake flasks. The strains were cultured at 30°C, 200 r.p.m. in minimal medium with 20 g l$^{-1}$ glucose. Cells from three independent colonies were used for three independent evolution series. Serial transfer was performed every day or every second day. For every transfer, the cell culture was diluted by a factor ranging from 1:6 to 1:10 into minimal glucose medium to give an initial OD$_{600}$ of about 0.1. After 40 days, the three populations were spread on YPD plates, and three clones were randomly picked from each line of evolution: sZJD-24-1A, sZJD-24-1B, and sZJD-24-1C from the first line, sZJD-24-2A, sZJD-24-2B, and sZJD-24-2C from the second line, and sZJD-24-3A, sZJD-24-3B, and sZJD-24-3C from the third line. The specific growth rates of these nine strains were determined in shake flasks with 20 g l$^{-1}$ minimal glucose medium.

**Genome sequencing**. The genomic DNA of the ALE strains was extracted by using the Blood & Cell culture DNA Kit. The quality of the genomic DNA was assessed by the Agilent 2100 Bioanalyzer according to the manufacturer's instructions. The genomic DNA of sZJD-24-1C and sZJD-24-2C failed to meet the sequencing quality requirement and was discarded. DNA from strains ZJD-24-1A, sZJD-24-1B, sZJD-24-2A, sZJD-24-2B, sZJD-24-3A, sZJD-24-3B, sZJD-24-3C, and parental strain sZJD-24 was sent for sequencing. The Illumina TruSeq Nano HT 96 was used for preparing libraries for genome sequencing, and whole-genome sequencing was performed on the Illumina NextSeq by using the 2 × 150 bp paired-end method according to manufacturer's manual. *S. cerevisiae* CEN.PK 113-7D was used as the reference genome (cenpk.tudelft.nl) for mapping the reads. Breseq-0.28.1 was used for detecting single nucleotide variants, insertions, and small deletions[37].

**Transcriptome analysis**. Cells were collected at OD$_{600}\approx 1$ and stored at −80°C before processing for RNA extraction. Total RNA was extracted by using the RNeasy Mini Kit. The quality of RNA samples was assessed by using the Agilent 2100 Bioanalyzer according to the manufacturer's instructions. The RNA samples were prepared by using the TruSeq RNA Stranded HT Sample Prep Kit (Illumina) and sequenced by NextSeq Series Mid-Output Kit (2 × 75) (Illumina). Reads were aligned on the yeast genome by using Bowtie2[38] and then further processed by SAMTools[39] and BEDTools[40] to count the number of reads aligning to each gene. Differential gene expression was analyzed by using the DEseq package in the R programming language[41]. Reporter analysis on GO terms and transcription factors was performed by using the Platform for Integrative Analysis of Omics (PIANO) R package[42]. Differential expression levels and p-adj values were used as input in reporter analysis. GO slim mapper analysis was conducted with the online tool at the SGD website (http://www.yeastgenome.org/cgi-bin/GO/goSlimMapper.pl).

**Data availability**. The RNA-Seq raw data of the reverse engineered strains and the control strain can be downloaded from the European Nucleotide Archive with the access number PRJEB23677 (https://www.ebi.ac.uk/ena/data/search?query = +PRJEB23677). The data that support the findings of this study are available within the article and its Supplementary Information file or available from the corresponding author upon reasonable request.

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

## Acknowledgements

This work was funded by the Novo Nordisk Foundation, the Knut and Alice Wallenberg Foundation, and the Strategic Research Council and Vetenskapsrådet. We appreciate the helpful discussions with Dina Petranovic, Yiming Zhang, Rui Pereira, Guodong Liu, Jianye Xia, Stefan Tippmann, Anastasia Krivoruchko, Florian David, Boyang Ji, and Jiufu Qin.

## Author contributions

Z.J.D. and J.N. conceived the study; Z.J.D. designed and performed all the experiments and analyzed the data; M.T.H. assisted with transcriptional data analysis; Z.J.D., M.T.H., Y.C., V.S., and J.N. wrote the manuscript.

## Additional information

**Competing interests:** The authors declare no competing interests.

