## [Peer Review File · Nature Communications]

Editorial Note: This manuscript has been previously reviewed at another journal that is not operating a transparent peer review scheme. This document only contains reviewer comments and rebuttal letters for versions considered at Nature Communications. Mentions of prior referee reports have been redacted.

PEER REVIEW FILE

Reviewer #1 (Not invited, absent during review by [REDACTED])

Reviewer #2 (Remarks to the Author):

The manuscript entitled »Global rewiring of cellular metabolism renders *Saccharomyces cerevisiae* Crabtree-negative« (NCOMMS-18-08523-T) describes genetic engineering of Crabtree positive *S. cerevisiae* yeast into a Crabtree negative. The authors conducted metabolic remodeling by engineering novel cytosolic acetyl-CoA synthetic pathway followed by the selection with adaptive laboratory evolution method to obtain strain with Crabtree negative phenotype. By integrated systems biology analyses they came to the conclusion that the mutation in the RNA polymerase II Mediator complex plays a significant role in metabolic changes. This is an interesting paper in large part because reducing aerobic ethanol production might open new perspectives for using *S. cerevisiae* as a commercial microorganism. The subject of the paper is highly original and certainly interesting to the scientists in various disciplines of biotechnology. The paper is clearly written, there is appropriate use of statistics and appropriate credit to previous work.

All the experiments in this paper were performed rigorously. The idea to increase growth rate of the pyruvate decarboxylase knock-down strain by redirecting pyruvate flow into acetyl-CoA was original and well conducted. Although the growth rate of initially engineered strain (ZJD-23) was too slow, fermentative production of various products especially acetate was successfully diminished. In continuation, they generated strains with higher growth rates comparable to that of the Crabtree positive yeasts with adaptive laboratory evolution (ALE) method. Concise whole genome sequencing of several evolved strains revealed mutations in MED2 or MED3 genes encoding tail module of RNA polymerase II Mediator complex and non-sense mutation in GPD1 encoding glycerol-3-phosphate dehydrogenase. By reverse engineering of MED2 and GPD1 targets in the initially generated strain, double mutant strain sZJD-28 was obtained. This final *S. cerevisiae* strain exhibited growth rate of 0,205 h⁻¹ that was comparable to that of the Crabtree-negative yeasts while the level of fermentation products was significantly reduced. Faster glucose consumption of the sZJD-28 strain was also observed (Fig. 2d).

Finally, the sZJD-28 strain and two other strains exhibiting increased growth rate were analyzed by conducting physiological characterization and gene ontology (GO). The authors found restricted glucose uptake and down-regulation of almost all glycolytic genes that interestingly

resulted in faster glucose consumption rate, higher glycolytic flux and respiration rate. They explain the phenomenon by a new balance between fermentation and respiration in the engineered *S. cerevisiae* that may lead to efficient carbon and electron flux in the cells which can support faster growth.

The authors adequately addressed all comments that I posed in my first review report. They conducted thorough physiological characterization of the sZJD-28 strain in respect to the wild type and parental strain. Accordingly they adequately re-wrote substantial part of the discussion. I find the present version of the manuscript sound enough to be published in the Nature Communications.

Dr. Matic Legiša

Reviewer #3 (Remarks to the Author):

Manuscript Review (Original)

Title: Global rewiring of cellular metabolism renders *Saccharomyces cerevisiae* Crabtree-negative

Journal: [REDACTED]

Authors: Zongjie Dai, Mingtao Huang, Yun Chen, Verena Siewers, and Jens Nielsen

Comments: While the manuscript presents a sound experimental study with interesting findings, in the opinion of this reviewer it lacks the novelty required for publication in [REDACTED]. The adaptive evolution approach has previously been utilized to isolate Pdc- *S. cerevisiae* strains with high growth rates in the presence of excess glucose (0.2 hr⁻¹) (DOI: 10.1128/AEM.70.1.159–166.2004). Here, the authors take a different approach for acetyl-CoA generation prior to adaptive evolution and while they identify alternative mutations leading to the increased growth rate (compared to the previous study), the maximum specific growth rates obtained here are only marginally improved (~10%) over the previously isolated strain. Furthermore, despite the authors stated motivation of improving product yields by abolishing the Crabtree effect, the only data presented using the identified mutations demonstrates a modest 20% increase in farnesene titer compared to the wild-type (Pdc+) background. The transcriptional analysis of strains harboring the identified mutation(s) provide the most intriguing findings, however some of the major gene classes identified in these strains were also identified in the previously isolated strain (e.g. hexose transporters). The identification and analysis of altered expression in the MED2 mutant strains represents the most novel findings, however in

my view demonstration of how engineering the Mediator complex can be used as a new strategy in metabolic engineering of yeast is required for this to warrant publication in [REDACTED] [REDACTED]. Additional specific comments are provided below:

1. The authors should explain their rationale for using 25 g/L glucose for fermentations with certain strains and 20 g/L for others. Does this impact the comparison between strains (for example, lines 114-116 comparing growth rates/glucose consumption/products of sZJD-28 to sZJD-23)?
2. Both farnesene and 3-hydroxypropanoate (3-HP) were initially used as metrics for acetyl-CoA derived products with strain sZJD-11, however for subsequent strains only farnesene was shown. How did 3-HP production compare in the evolved and re-engineered strains?
3. The authors state (line 105) that MED2, MED3, GPD1, HXK2 and SIW14 were chosen as reverse engineering targets for further evaluation, however results are only shown/discussed for MED2 and GPD1. Please elaborate on the results with mutations to the other genes.
4. The manuscript has numerous grammatical errors and many sentences in which the syntax makes it difficult to understand the authors rationale/discussion. The text should be thoroughly edited.

Manuscript Review (Revision)

Title: Global rewiring of cellular metabolism renders *Saccharomyces cerevisiae* Crabtree-negative

Authors: Zongjie Dai, Mingtao Huang, Yun Chen, Verena Siewers, and Jens Nielsen

Comments: The reviewer thanks the authors for their modifications to the original manuscript which in my view results in an improved version of the manuscript. The expanded discussion on the transcriptional analysis of strains harboring the identified mutation(s) provides the reader with a much-improved understanding of the authors overall hypothesis of the effect of the identified mutation(s). The manuscript should be considered for publication in Nature Communications. Please note that numerous grammatical errors are still present in the manuscript which should go through a rigorous edit prior to publication.

Reviewer #4 (Remarks to the Author):

Previous review of this manuscript by this reviewer :

This is an elegant study in which, by genetic and ALE means, the authors increase the growth yield of a Crabtree positive yeast and thus claim they turned it into a Crabtree negative one. This is a well conceived and rigorous study. The data presented are robust. However, for the authors claim to be supported by experimental data it is mandatory that the authors assess the cellular respiratory rate which is one of the three features of the yeast Crabtree effect (-1-increase in growth rate,-2- decrease in cell respiration and -3-increase in glycolysis flux). Indeed upon induction of this effect, respiratory rate is highly decreased. If their strain is Crabtree negative, no decrease in the cellular respiratory rate should be observed in high glucose growth conditions. Moreover, this reviewer is surprised by the fact that in figure 3 (transcriptional analysis), there is no analysis of OXPHOS genes that could support the authors claim (i.e. no OXPHOS repression in high glucose conditions in their "Crabtree negative" strain).

The authors answers to this reviewer's comments were satisfactory. It would however be useful to the reader if the authors specified in their material and methods section the methods used to assess QO₂ and QCO₂.

Responses to Reviewers:

REVIEWERS' COMMENTS:

Reviewer #1 (Not invited, absent during review by [REDACTED])

Reviewer #2 (Remarks to the Author):

The manuscript entitled »Global rewiring of cellular metabolism renders *Saccharomyces cerevisiae* Crabtree-negative« (NCOMMS-18-08523-T) describes genetic engineering of Crabtree positive *S. cerevisiae* yeast into a Crabtree negative. The authors conducted metabolic remodeling by engineering novel cytosolic acetyl-CoA synthetic pathway followed by the selection with adaptive laboratory evolution method to obtain strain with Crabtree negative phenotype. By integrated systems biology analyses they came to the conclusion that the mutation in the RNA polymerase II Mediator complex plays a significant role in metabolic changes. This is an interesting paper in large part because reducing aerobic ethanol production might open new perspectives for using *S. cerevisiae* as a commercial microorganism. The subject of the paper is highly original and certainly interesting to the scientists in various disciplines of biotechnology.

The paper is clearly written, there is appropriate use of statistics and appropriate credit to previous work.

All the experiments in this paper were performed rigorously. The idea to increase growth rate of the pyruvate decarboxylase knock-down strain by redirecting pyruvate flow into acetyl-CoA was original and well conducted. Although the growth rate of initially engineered strain (ZJD-23) was too slow, fermentative production of various products especially acetate was successfully diminished. In continuation, they generated strains with higher growth rates comparable to that of the Crabtree positive yeasts with adaptive laboratory evolution (ALE) method. Concise whole genome sequencing of several evolved strains revealed mutations in MED2 or MED3 genes encoding tail module of RNA polymerase II Mediator complex and non-sense mutation in GPD1 encoding glycerol-3-phosphate dehydrogenase. By reverse engineering of MED2 and GPD1 targets in the initially generated strain, double mutant strain sZJD-28 was obtained. This final *S. cerevisiae* stain exhibited growth rate of 0,205 h⁻¹ that was

comparable to that of the Crabtree-negative yeasts while the level of fermentation products was significantly reduced. Faster glucose consumption of the sZJD-28 strain was also observed (Fig. 2d). Finally, the sZJD-28 strain and two other strains exhibiting increased growth rate were analyzed by conducting physiological characterization and gene ontology (GO). The authors found restricted glucose uptake and down-regulation of almost all glycolytic genes that interestingly resulted in faster glucose consumption rate, higher glycolytic flux and respiration rate. They explain the phenomenon by a new balance between fermentation and respiration in the engineered *S. cerevisiae* that may lead to efficient carbon and electron flux in the cells which can support faster growth.

The authors adequately addressed all comments that I posed in my first review report. They

conducted thorough physiological characterization of the sZJD-28 strain in respect to the wild type and parental strain. Accordingly they adequately re-wrote substantial part of the discussion. I find the present version of the manuscript sound enough to be published in the Nature Communications.

Dr. Matic Legiša

Response: Thank you very much for your suggestions, comments and approval.

Reviewer #3 (Remarks to the Author):

Manuscript Review (Original)

Title: Global rewiring of cellular metabolism renders *Saccharomyces cerevisiae* Crabtree-negative

Journal: [REDACTED]

Authors: Zongjie Dai, Mingtao Huang, Yun Chen, Verena Siewers, and Jens Nielsen

Comments: While the manuscript presents a sound experimental study with interesting findings, in the opinion of this reviewer it lacks the novelty required for publication in [REDACTED]. The adaptive evolution approach has previously been utilized to isolate Pdc- *S. cerevisiae* strains with high growth rates in the presence of excess glucose (0.2 hr⁻¹) (DOI: 10.1128/AEM.70.1.159–166.2004). Here, the authors take a different approach for acetyl-CoA generation prior to adaptive evolution and while they identify alternative mutations leading to the increased growth rate (compared to the previous study), the maximum specific growth rates obtained here are only marginally improved (~10%) over the previously isolated strain. Furthermore, despite the authors stated motivation of improving product yields by abolishing the Crabtree effect, the only data presented using the identified mutations demonstrates a modest 20% increase in farnesene titer compared to the wild-type (Pdc+)

background. The transcriptional analysis of strains harboring the identified mutation(s) provide the most intriguing findings, however some of the major gene classes identified in these strains were also identified in the previously isolated strain (e.g. hexose transporters). The identification and analysis of altered expression in the MED2 mutant strains represents the most novel findings, however in my view demonstration of how engineering the Mediator complex can be used as a new strategy in metabolic engineering of yeast is required for this to warrant publication in [REDACTED]. Additional specific comments are provided below:

1. The authors should explain their rationale for using 25 g/L glucose for fermentations with certain strains and 20 g/L for others. Does this impact the comparison between strains (for example, lines 114-116 comparing growth rates/glucose consumption/products of sZJD-28 to sZJD-23)?
2. Both farnesene and 3-hydroxypropanoate (3-HP) were initially used as metrics for acetyl-CoA derived products with strain sZJD-11, however for subsequent strains only farnesene was shown. How did 3-HP production compare in the evolved and re-engineered strains?
3. The authors state (line 105) that MED2, MED3, GPD1, HXK2 and SIW14 were chosen as reverse engineering targets for further evaluation, however results are only shown/discussed for MED2 and

GPD1. Please elaborate on the results with mutations to the other genes.

4. The manuscript has numerous grammatical errors and many sentences in which the syntax makes it difficult to understand the authors rationale/discussion. The text should be thoroughly edited.

Manuscript Review (Revision)

Title: Global rewiring of cellular metabolism renders *Saccharomyces cerevisiae* Crabtree-negative

Authors: Zongjie Dai, Mingtao Huang, Yun Chen, Verena Siewers, and Jens Nielsen

Comments: The reviewer thanks the authors for their modifications to the original manuscript which in my view results in an improved version of the manuscript. The expanded discussion on the transcriptional analysis of strains harboring the identified mutation(s) provides the reader with a much-improved understanding of the authors overall hypothesis of the effect of the identified mutation(s). The manuscript should be considered for publication in Nature Communications. Please note that numerous grammatical errors are still present in the manuscript which should go through a rigorous edit prior to publication.

Response: Thanks a lot for your comments and approval.

Reviewer #4 (Remarks to the Author):

Previous review of this manuscript by this reviewer :

This is an elegant study in which, by genetic and ALE means, the authors increase the growth yield of a Crabtree positive yeast and thus claim they turned it into a Crabtree negative one. This is a well conceived and rigorous study. The data presented are robust. However, for the authors claim to be supported by experimental data it is mandatory that the authors assess the cellular respiratory rate which is one of the three features of the yeast Crabtree effect (-1-increase in growth rate,-2-decrease in cell respiration and -3-increase in glycolysis flux). Indeed upon induction of this effect, respiratory rate is highly decreased. If their strain is Crabtree negative, no decrease in the cellular respiratory rate should be observed in high glucose growth conditions. Moreover, this reviewer is surprised by the fact that in figure 3 (transcriptional analysis), there is no analysis of OXPHOS genes that could support the authors claim (i.e. no OXPHOS repression in high glucose conditions in their "Crabtree negative" strain).

The authors answers to this reviewer's comments were satisfactory. It would however be useful to the reader if the authors specified in their material and methods section the methods used to assess QO₂ and QCO₂.

Response: Thank you very much for your suggestions. We added the calculation method for qO₂ and qCO₂ in Methods section.